# Sanguinarine Inhibits Mono- and Dual-Species Biofilm Formation by *Candida albicans* and *Staphylococcus aureus* and Induces Mature Hypha Transition of *C. albicans*

**DOI:** 10.3390/ph13010013

**Published:** 2020-01-13

**Authors:** Weidong Qian, Wenjing Wang, Jianing Zhang, Miao Liu, Yuting Fu, Xiang Li, Ting Wang, Yongdong Li

**Affiliations:** 1School of Food and Biological Engineering, Shaanxi University of Science and Technology, Xi’an 710021, China; qianweidong@sust.edu.cn (W.Q.); 201504040311@sust.edu.cn (W.W.); 1804070@sust.edu.cn (J.Z.); 1804065@sust.edu.cn (M.L.); 201504040401@sust.edu.cn (Y.F.); lixiang@sust.edu.cn (X.L.); 2Ningbo Municipal Center for Disease Control and Prevention, Ningbo 315010, China

**Keywords:** *Candida albicans*, *Staphylococcus aureus*, sanguinarine, dual-species, antibiofilm activity

## Abstract

Previous studies have reported that sanguinarine possesses inhibitory activities against several microorganisms, but its effects on mono- and dual-species biofilms of *C. albicans* and *S. aureus* have not been fully elucidated. In this study, we aimed to evaluate the efficacy of sanguinarine for mono- and dual-species biofilms and explore its ability to induce the hypha-to-yeast transition of *C. albicans*. The results showed that the minimum inhibitory concentration (MIC) and minimum biofilm inhibitory concentration (MBIC90) of sanguinarine against *C. albicans* and *S. aureus* mono-species biofilms was 4, and 2 μg/mL, respectively, while the MIC and MBIC90 of sanguinarine against dual-species biofilms was 8, and 4 μg/mL, respectively. The decrease in the levels of matrix component and tolerance to antibiotics of sanguinarine-treated mono- and dual-species biofilms was revealed by confocal laser scanning microscopy combined with fluorescent dyes, and the gatifloxacin diffusion assay, respectively. Meanwhile, sanguinarine at 128 and 256 μg/mL could efficiently eradicate the preformed 24-h biofilms by mono- and dual-species, respectively. Moreover, sanguinarine at 8 μg/mL could result in the transition of *C. albicans* from the mature hypha form to the unicellular yeast form. Hence, this study provides useful information for the development of new agents to combat mono- and dual-species biofilm-associated infections, caused by *C. albicans* and *S. aureus*.

## 1. Introduction

*Candida albicans* (*C. albicans*) is a frequent commensal colonizer of the human skin and mucosal flora [1]. However, *C. albicans* is an opportunistic fungal pathogen, and can cause infections (candidiasis or thrush) in humans and other animals, where it is estimated that more than 70% of women experience Candida infection at least once in their lifetime [2,3]. *C. albicans* also readily develops biofilms on the surface on in-dwelling biomedical materials and mucosal tissues, which serve as a rich reservoir of infection, and can result in severe systemic infections because of their heightened resistance to antibiotics [4]. Critically, studies have suggested that in a natural, clinical, and industrial environment, microbes are rarely present as single-species communities, but instead the vast majority exist as part of complex multi-species consortia, where a mutually beneficial interaction between members of interspecific species may develop [5]. Likewise, an estimated 27% to 56% of nosocomial *C. albicans* bloodstream infections may be defined as polymicrobial [6]. Importantly, *Staphylococcus aureus* (*S. aureus*), which is recognized as one of the most significant disease-causing bacteria in humans, was found to be the third most commonly co-isolated species in conjunction with *C. albicans* [7,8]. Interestingly, the combined effect of *C. albicans* and *S. aureus* led to synergism and enhanced mortality in mice [9].

*C. albicans* and *S. aureus* specifically are acknowledged as leading opportunistic fungal, and bacterial pathogens, respectively, mainly due to their ability to form biofilms on catheters and indwelling medical devices [10]. In the presence of *C. albicans*, *S. aureus* can form a substantial polymicrobial biofilm, where *S. aureus* cells are interspersed within the filamentous fungal network, harnessing *C. albicans* as the physical scaffold [11,12]. Biofilm formation of mixed species represents a protected mode of microbial growth that, not only renders cells able to withstand hostile environments, but also to promote and colonize new niches by dispersal of microorganisms from the microbial clusters [13]. Several studies have demonstrated the interactions in dual species biofilms between *C. albicans* and *S. aureus*, in which the biomass, bio-volume, and thickness of mixed-species biofilms were significantly higher compared with mono-species biofilms, thereby, directly decreasing the penetration of the antibiotic through the biofilm matrix, and thus, resulting in strong resistance to antibiotics, as well as higher expression levels of most virulence factors [14]. To prevent and combat such nosocomial medical device-related infections, caused by biofilms formed by mono- and dual-species of *C. albicans* and *S. aureus*, there is an urgent need for new effective antimicrobial and anti-biofilm agents.

Natural extracts have drawn much attention for their therapeutic effects in the treatment and prevention of various diseases, and as a potential source of promising antimicrobial agents due to their high biocompatibility and low toxicity. Sanguinarine, a benzophenanthridine alkaloid extracted from plants belonging to the family Papaveracea, elicits a wide range of biological effects, including anti-inflammatory and anti-cancer properties [15,16,17]. Interestingly, sanguinarine was found to be a potent inhibitor of *S. aureus*, *Escherichia coli* and *Psoroptes cuniculi*, and it was found to have anti-biofilm activity against *C. albicans* [18,19,20]. However, the inhibitory and scavenging effects of sanguinarine on *S. aureus* mono-species, and dual-species biofilms of *C. albicans* and *S. aureus*, as well as its influence on mature hypha of *C. albicans* remains to be fully elucidated. The objective of the present study was to evaluate the anti-biofilm and mature biofilm eradication activities of sanguinarine against *S. aureus* mono-species, and dual-species biofilms of *C. albicans* and *S. aureus*, and also explore its effect on the mature hypha switch of *C. albicans*.

## 2. Results

### 2.1. Minimum Inhibitory Concentration (MIC) and Minimum Biofilm Inhibitory Concentration (MBIC90) of Sanguinarine Against C. albicans SC5314 and S. aureus CMCC26003

*C. albicans* SC5314 and *S. aureus* CMCC26003 mono- and dual-cultures were tested for susceptibility to sanguinarine. Sanguinarine exhibited potent antimicrobial activities to mono- and dual-cultures. As presented in Table 1, sanguinarine was found to be effective with low MIC and MBIC90 of 4 μg/mL and 2 μg/mL against both *C. albicans* SC5314, and *S. aureus* CMCC26003 single cultures, respectively. In addition, the MIC and MBIC90 of sanguinarine against mixed cultures was 8 μg/mL and 4 μg/mL.

### 2.2. Sanguinarine Suppresses the Biofilm Formation of Mono- and Dual-Species

As demonstrated in Figure 1, the addition of sanguinarine to mono- or dual-cultures inhibited biofilm formation in a dose-dependent manner. Specifically, sanguinarine (0.5 μg) demonstrated a significant anti-biofilm activity against *C. albicans* SC5314 at the concentration of 1/8 MIC (*p* < 0.05). Similarly, 1/8 MIC of sanguinarine (0.5 and 1 μg) significantly suppressed the biofilm formation of *S. aureus* CMCC26003 mono-species and dual-species (*p* < 0.05). In addition, following sanguinarine treatment with 1/2 MIC or above, biofilms of both mono- and dual-species were characterized by a very low bio-volume, and significant differences were observed between treated and untreated biofilms (*p* < 0.001). Overall, the results showed that sanguinarine had an excellent inhibitory effect on the biofilm formation of mono- and dual-species of *C. albicans* SC5314 and *S. aureus* CMCC26003, and MBIC90 of sanguinarine against *C. albicans* SC5314 and *S. aureus* CMCC26003 mono-species biofilms was 2 μg/mL, whereas, the MBIC90 of sanguinarine against mixed biofilms was 4 μg/mL (Table 1).

### 2.3. Sub-MIC Sanguinarine Changed the Biofilm Structure of Mono- and Dual-Species

As shown in Figure 2A, field emission scanning electron microscopy (FESEM; Nova Nano SEM-450, FEI, Hillsboro, OR, USA) images showed that mono- and dual-species cells gave rise to dense biofilms on the cover-slides in the absence of sanguinarine. Under treatment with sanguinarine at 1/8 MIC, the cell density of mono- and dual-species decreased slightly, suggesting that the effect of sanguinarine at low concentration on biofilm structure, was not remarkable. In contrast, in the case of the ¼ MIC sanguinarine-treated group, the mono-species biofilm cells were aggregated in distinct clusters, and had low aggregation propensity compared with the untreated group. For dual-species biofilms, cells treated with sanguinarine at ¼ MIC presented an attached *C. albicans* cell on the surface, to which *S. aureus* grape-like cell clusters clung. Obviously, in the treated group with sanguinarine at ½ MIC, the *C. albicans* SC5314 mono- and dual-species biofilm structures were missing and only individual cells could be seen, and for *S. aureus* CMCC26003 biofilms, amorphous and scattered cells were observed.

Similarly, confocal laser scanning microscopy (CLSM; Zeiss LSM 880 with Airyscan) images revealed biofilm morphologies and supported the FESEM results (Figure 2B). In *C. albicans* SC5314 mono-species biofilm, a large number of flake-like biofilms were observed in the untreated *C. albicans* SC5314 pure-culture group. After treatment with sanguinarine, the biofilm biomass was significantly reduced, and appeared looser and more dispersed with increased sanguinarine concentration. Moreover, *S. aureus* CMCC26003 mono-species biofilms were much denser compared to *C. albicans* SC5314 biofilms. In the presence of sanguinarine at 1/4 and ½ MIC, the density of *S. aureus* CMCC26003 cells was overtly attenuated, and the less dense biofilm with the bubble-like structure was observed. Similarly, in dual-species biofilms of *C. albicans* SC5314 and *S. aureus* CMCC26003, mixed-culture biofilms, treated with sanguinarine at 1/8 MIC formed a luxuriant biofilm, while the biofilm biomass remarkably decreased in the ½ MIC group as compared with the ¼ MIC group. These results showed that sanguinarine at ¼ and ½ MIC could effectively inhibit the dual-species biofilm formation of *C. albicans* SC5314 and *S. aureus* CMCC26003 within 24 h.

### 2.4. Sanguinarine Decreases the Biofilm Formation of C. albicans and S. aureus Mono- and Dual-Species by Mediating Extracellular Protein Levels in a Similar Manner

In *C. albicans* biofilms, proteins found in the biofilm matrix include a few predicted to form part of the secretome (mostly glycoproteins), but also many secretion-signal-less proteins [21]. Similarly, in *S. aureus* biofilms, the proteinaceous matrix is principally composed of cytoplasmic proteins [22]. To visualize the inhibitory effect of sanguinarine on extracellular protein levels within mono- and dual-species biofilms by CLSM, proteinaceous components and cells within biofilms formed by mono- and dual-species of *C. albicans* SC5314 and *S. aureus* CMCC26003 were stained with Film Tracer SYPRO Ruby (SYPRO Ruby; Invitrogen, Thermo Fisher Scientific, Waltham, MA, USA), and STYO 9, respectively. SYPRO Ruby, which can label most classes of proteins, including glycoproteins, phosphoproteins, lipoproteins, calcium binding proteins and fibrillar proteins. According to Figure 3A, in treated biofilms of *C. albicans* SC5314 exposed to ¼ MIC, or ½ MIC of sanguinarine, the protein levels of biofilms remarkably declined compared to the untreated group. Similarly, in *S. aureus* CMCC26003 mono-species biofilms, a large number of extracellular proteins were seen and evenly embedded in the biofilm in the mosaic-like pattern in the control group, while extracellular proteins within biofilms were shown to have greatly reduced, or were even invisible, upon exposure to ¼ MIC or ½ MIC of sanguinarine. Moreover, the level of extracellular proteins was similarly decreased in dual-species biofilms when treated with ¼ MIC of sanguinarine, and a lower total extracellular protein level was observed under ½ MIC of sanguinarine.

To further assess the relative levels of extracellular proteins per number of cells within biofilms formed by mono- and dual-species of *C. albicans* SC5314 and *S. aureus* CMCC26003, we calculated the ratio of protein to cell fluorescence. As displayed in Figure 3B, the average ratios of extracellular protein to cell fluorescence measured in untreated *C. albicans* SC5314 and *S. aureus* CMCC26003 mono-species biofilms were statistically higher than those in biofilms of ¼ MIC treated group (100% and 100% versus 26.3% ± 1.5% and 13.3% ± 0.6%, respectively; *p* < 0.001). Similarly, the average ratio of extracellular proteins to cell fluorescence determined in untreated dual-species biofilms was statistically more than that in biofilms of ¼ MIC treated group (100.0% versus 17.6% ± 0.5%, respectively; *p* < 0.001). The results indicate that the extracellular protein content, within SC5314 and *S. aureus* CMCC26003 mono- and dual-species biofilms, decreased as the sanguinarine concentration increased.

### 2.5. Sanguinarine Reduced the Biofilm Formation of S. aureus Mono-Species and Dual-Species by Mediating Extracellular Polysaccharide Levels

*C. albicans* biofilm matrix is complex, with major polysaccharide constituents being α-mannan, β-(1,6)-glucan and β-(1,3)-glucan [23]. Similarly, a primary component of the *S. aureus* biofilm matrix was polysaccharide intercellular adhesin, which is also known as poly-*N*-acetyl-β-(1,6)-glucosamine [24]. To evaluate the inhibitory effect of sanguinarine on extracellular polysaccharides levels of mono- and dual-species biofilms, extracellular polysaccharides and cells within biofilms were labelled with wheat germ agglutinin (WGA; green), and FM 4-64 (red), respectively. WGA can bind to *N*-acetyl-d-glucosamine. As shown in Figure 4A, in *C. albicans* SC5314 mono-species and dual-species biofilms, the content of extracellular polysaccharides decreased with increasing concentrations of sanguinarine, and a significant reduction was observed with ¼ MIC of sanguinarine. In contrast, in *S. aureus* CMCC26003 mono-species biofilms, a similar reduction of extracellular polysaccharides was seen when exposed to 1/8 MIC of sanguinarine. Moreover, in dual-species biofilms, sanguinarine reduced the extracellular polysaccharide levels in a dose-dependent manner, where both yellow and red fluorescence were linearly related with the concentration of sanguinarine, and the overlapping red and green signals yielded an unambiguous bright yellow signal, suggesting that extracellular polysaccharides were tightly bound to the surface of dual-culture cells.

To further examine the relative amounts of extracellular polysaccharides per number of cells in biofilms, formed by mono- and dual-species of *C. albicans* SC5314 and *S. aureus* CMCC26003, we measured the ratios of polysaccharide to cell fluorescence. As demonstrated in Figure 4B, the relative fluorescence intensities of extracellular polysaccharide measured in untreated *C. albicans* SC5314 mono-species and dual-species biofilms were statistically higher than those in biofilms of ¼ MIC treated group (100% and 100% versus 23.0% ± 1.6% and 24.3% ± 0.6%, respectively; *p* < 0.001), respectively. In contrast, there was a statistically significant difference in the relative fluorescence value of extracellular polysaccharide between the untreated and 1/8 MIC-treated groups (100% versus 35.0% ± 1.7%, respectively; *p* < 0.001) for *S. aureus* CMCC26003 mono-species biofilms.

### 2.6. Sanguinarine Inhibits the Biofilm Formation of Dual Species by Reducing Extracellular DNA Levels

eDNA is a crucial component of *C. albicans* and *S. aureus* mature biofilms as it contributes to biofilm structural integrity and stability [25]. To investigate how extracellular DNA (eDNA) contributed to the observed decrease in mono- and dual-species of *C. albicans* SC5314 and *S. aureus* CMCC26003 biofilms, after treatment with sanguinarine by CLSM, eDNA was stained with a red fluorescent membrane-intact impermeable DNA-binding stain propidium iodide (PI). As shown in Figure 5A, we observed a decrease in red signal with the rise of sanguinarine concentration for dual-species biofilms, indicating that sanguinarine reduced the eDNA quantity for these biofilms in a dose-dependent manner. In *C. albicans* SC5314 mono-species biofilms, red fluorescence signals were less visible regardless of whether sanguinarine was present or not, revealing that the matrix of *C. albicans* SC5314 biofilms contains a small amount of eDNA, which has little effect on the biofilm formation of *C. albicans* SC5314 biofilms. A similar trend was found in dual-species biofilms, where the red signal intensity increased with sanguinarine concentration. In contrast, in *S. aureus* CMCC26003 mono-species biofilms, a number of scattered and bulk yellow colorations caused by overlaps of green (microbial cells) and red (eDNA) fluorescence signals were observed, but there were minor changes in eDNA levels in both the control and sanguinarine-treated groups.

To further investigate the relative abundance of eDNA per number of cells in biofilms formed by mono- and dual-species of *C. albicans* SC5314 and *S. aureus* CMCC26003, the ratio of eDNA to cell fluorescence was examined. Imaging measurements demonstrated that the relative fluorescence intensities of eDNA of untreated *C. albicans* SC5314 mono-species and dual-species biofilms were statistically higher than those of biofilms of ¼ MIC treated group (100% and 100% versus 21.6% ± 2.1%, and 23.0% ± 1.7%, respectively; *p* < 0.001) (Figure 5B). Similarly, the relative fluorescence intensity of eDNA in untreated biofilms of *S. aureus* CMCC26003 demonstrated statistically a little higher than that in biofilms of ½ MIC treated group (100% versus 17.0% ± 1.0%, respectively; *p* < 0.001) (Figure 5B). The results indicate that the eDNA decrease contributes to a great extent to sanguinarine-reduced biofilms formed by dual-species, while this situation was not applicable for *C. albicans* SC5314 and *S. aureus* CMCC26003 mono-species biofilms in response to sanguinarine.

### 2.7. Sanguinarine Treatment Reduces Mono- and Dual-Species Biofilm Tolerance to Gatifloxacin

To determine the tolerance of mono- and dual-species biofilms, treated with sanguinarine to antibiotics, CLSM was used in combination with gatifloxacin with intrinsic fluorescence. According to Figure 6, there was a significant increase in gatifloxacin diffusion within mono- and dual-species biofilms, formed in the presence of sanguinarine. In untreated mono- and dual-species biofilms, gatifloxacin was disposed along the outer periphery of the biofilm with minimal-to-no penetration into the biofilm. Similarly, in the presence of sanguinarine at 1/8 MIC, blue signals were detected less, and only a small amount of gatifloxacin penetrated sporadically into the matrix. In contrast, under exposure to sanguinarine at ¼ MIC, gatifloxacin penetration within the biofilms was significantly increased for mono- and dual-species biofilms, and gatifloxacin was diffused throughout the biofilm matrix. In particular, in the presence of ½ MIC sanguinarine, gatifloxacin had obviously diffused throughout the mono- and dual-species biofilm matrix and reached the basal layers.

### 2.8. Sanguinarine Efficiently Eradicates Mature Biofilms of Mono- and Dual-Species

The eradication activity of sanguinarine on mature biofilms of mono- and dual-species was examined by the crystal violet assay. As shown in Figure 7A, all mature biofilms of mono- and dual-species were almost eradicated under the condition of sanguinarine at 32 MIC. *C. albicans* SC5314 and *S. aureus* CMCC26003 mono-species biofilms, treated with 8 MIC sanguinarine, showed a significant reduction in biomass (*p* < 0.01), and a greater decrease in biofilm biomass was observed (*p* < 0.001) when exposed to sanguinarine at 16 MIC or 32 MIC. A similar trend was found in dual-species biofilms, where the biofilm biomass was remarkably decreased at a high dose of sanguinarine (32 MIC) (*p* < 0.001).

The eradication effect of sanguinarine on mono- and dual-species biofilms was further analyzed by FESEM. As seen in Figure 7B, in the absence of sanguinarine, mature biofilms of mono- and dual-species were composed of structural motifs consisting of dense, and ordered, honeycomb-like chambers. In contrast, the treatment of *C. albicans* SC5314 mono-species and dual-species biofilms with 32 MIC of sanguinarine resulted in a striking, almost complete disappearance of biofilm architecture. Similarly, in *S. aureus* CMCC26003 mono-species biofilms, treated with the 32 MIC group, the biofilm structure was visibly sparse, and the biofilm structure largely disappeared.

### 2.9. Sanguinarine Results in the Hypha-to-Yeast Transition in C. albicans

To examine the effect of sanguinarine on the hypha-to-yeast morphological transition of *C. albicans* SC5314, *C. albicans* cells were grown in Roswell Park Memorial Institute 1640 medium (RPMI 1640; Invitrogen, Grand Island, NY) supplemented with 10% fetal bovine serum (FBS), which is known to induce morphological transition. As shown in Figure 8, in the absence of sanguinarine, *C. albicans* produced elongated and regular hyphal cells. Mature *C. albicans* hypha exposed to ½ MIC of sanguinarine for 5 h transformed into shorter hyphae. Moreover, a significant population of yeast was observed when treated with sanguinarine at MIC. Notably, under treatment with sanguinarine at 2 MIC, *C. albicans* cells were completely devoid of hyphae and remained in yeast form.

## 3. Discussion

*C. albicans* and *S. aureus* can be co-isolated from a number of infections and exhibit enhanced disease severity and morbidity compared to single species infection [26]. The synergy between *C. albicans* and *S. aureus* contributes to the recalcitrance of polymicrobial biofilm communities, which has proven to be very resistant to a range of antibiotics, highlighting the emergent need for new antibiofilm agents [27]. Here, we first showed that sanguinarine imparted a cell growth-inhibitory response in mono- and dual-species planktic cells of *C. albicans* SC5314 and *S. aureus* CMCC26003. The results of the present study showed that sanguinarine gave potent antifungal activity against *C. albicans* SC5314 and *S. aureus* CMCC26003 single culture, with a MIC of 4 μg/mL, whereas, dual cultures of sanguinarine showed a MIC of 8 μg/mL. These results are consistent with previous reports indicating that sanguinarine exhibited potent antibacterial activity against both Gram-positive and Gram-negative bacteria, and antifungal activity against *Candida* and *dermatophytes* [19].

Moreover, we further explored the antibiofilm properties of sanguinarine against mono- and dual-species of *C. albicans* SC5314 and *S. aureus* CMCC26003 by CV assay, FESEM and CLSM. The results showed that the biofilm formation of *C. albicans* SC5314 and *S. aureus* CMCC26003 single species were almost completely suppressed by sanguinarine, with a MBIC of 2 μg/mL, while biofilm formation of dual species was inhibited by sanguinarine, with a MBIC90 of 4 μg/mL. In particular, the MBIC90 of sanguinarine against single and dual species of *C. albicans* SC5314 and *S. aureus* CMCC26003 was less in the present study than that of previous reports. Shin and Eom had previously shown that zerumbone remarkably suppressed mono and dual-species biofilms formed by *C. albicans* ATCC 14053 and *S. aureus* ATCC 29213 at 64 μg/mL [28]. Similarly, She et al. reported that auranofin could remarkably inhibit *S. aureus* and *C. albicans* ATCC14053 mono and dual biofilm formation in vitro at 4 μg/mL [29]. It may, therefore, be concluded from the above reports that sanguinarine had robust antibiofilm activities against mono- and dual-species of *C. albicans* and *S. aureus*. In addition, *C. albicans* and *C. albicans*/*S. aureus* preformed biofilms were less susceptible to conventional antifungal fluconazole treatment, in which fluconazole with more than 512 μg/mL can only eradicate 80% of performed *C. albicans* biofilms, and 6.05% of performed *C. albicans* and *S. aureus* dual-species biofilms, respectively [30]. Interestingly, 128 μg/mL sanguinarine could efficiently eradicate 91.8% of performed *C. albicans* SC5314 biofilms, and 91.1% of performed *C. albicans* SC5314 and *S. aureus* CMCC26003 dual-species biofilms, confirming its ability to reduce mature biofilms.

Next, the profile of extracellular proteins, polysaccharides and eDNA levels within biofilms formed by mono- and dual-culture of *C. albicans* and *S. aureus* after exposure to sanguinarine at sub-MIC was examined by CLSM, combined with five different fluorescent dyes, which were applied to differentiate bacterial cells from proteins, polysaccharides, and eDNA, respectively. The results obtained here suggested that sanguinarine reduced the biofilm formation of *S. aureus* single species mainly by mediating polysaccharides and eDNA levels, while for *C. albicans* single species and dual species, sanguinarine inhibited biofilm formation mainly by reducing protein and polysaccharide levels. The results were in line with previous reports that showed that the main components of *S. aureus* biofilms are composed of extracellular polysaccharides and eDNA, while the *C. albicans* biofilm matrix predominantly consists of extracellular proteins (55%), polysaccharides (25%), lipids (15%), and eDNA (5%) [31,32]. In agreement, for dual culture of *C. albicans* and *S. aureus*, sanguinarine reduced biofilm formation by synchronously reducing extracellular proteins, polysaccharides, and eDNA levels in a dose-dependent manner.

Meanwhile, the tolerance of biofilms to antibiotics is one of the specific factors precipitating disease emergence [33]. Here, we investigated the tolerance of sanguinarine-treated biofilms of mono- and dual-culture of *C. albicans* SC5314 and *S. aureus* CMCC26003 to antibiotics. To this end, gatifloxacin with intrinsic fluorescence, combined with CLSM, was employed to investigate the antibiotic diffusion through impaired biofilms. The results showed that 1/8 MIC sanguinarine has virtually no effect on the gatifloxacin penetration within biofilms. Whereas, gatifloxacin diffusion, in both mono- and dual-species biofilms, were enhanced markedly when treated with sanguinarine at high concentrations of ½ MIC, revealing that the physical integrity and biomass of the mono- and dual-species biofilms was disrupted and reduced by sanguinarine, thereby significantly enhancing the osmolality of the biofilms.

Microbial cells within biofilms have increased (up to 1000-fold higher) resistance to antimicrobials [34]. We also investigated the eradication activity of sanguinarine against the mature biofilms formed by mono- and dual-species of *C. albicans* SC5314 and *S. aureus* CMCC26003, followed by verifying the antibiofilm activities of sanguinarine. Here, sanguinarine could efficiently eradicate almost all 24-h mature biofilms formed by *C. albicans* SC5314 single-species and dual-species at concentrations of 128, and 256 μg/mL, respectively, and ≥90% 24-h mature biofilms of *S. aureus* CMCC26003 at the concentration of 128 μg/mL. In contrast, Shin and Eom reported that treatment with high concentration of 500 μg/mL of zerumbone resulted in a dramatic elimination of preformed *C. albicans* ATCC 14053 biofilm by 96.3% [29]. On the other hand, zerumbone only reduced preformed *S. aureus* ATCC 29213 and dual-species biofilms by 49% and 39.4% at concentrations of 500 μg/mL, suggesting that sanguinarine was more effective in eradicating mono- and dual-species of *C. albicans* and *S. aureus* preformed biofilms than zerumbone.

Both *C. albicans* morphologies, yeast, and hyphae, can adhere to the apical surface of cultured enterocytes, but hyphal growth forms appear to have a dominant role in mediating the penetration of monolayers of intestinal cells by disseminating *C. albicans* to the host tissues. Thus, the induction of the transition of *C. albicans* from hyphae to yeast forms is a prime example of reducing the virulence of *C. albicans* to the target [35]. In the current study, the reversal effect of sanguinarine on mature hyphae of *C. albicans* SC5314 was examined. Mature hyphae of *C. albicans* SC5314, in the presence of 8 μg/mL of sanguinarine, were shortened and obviously converted into the yeast state, which can be attributed to the reversible morphological plasticity of *C. albicans* SC5314 between yeast and hyphal forms in response to sanguinarine. Recently, Zhong et al. assessed the toxicity of sanguinarine using human umbilical vein endothelial cells (HUVECs) and found that the half-maximal (50%) inhibitory concentration (IC50) of sanguinarine on HUVECs was 7.8 μg/mL and sanguinarine at the concentration over than 12.8 μg/mL began to induce observed toxic effect toward the mammalian cells [19]. The results indicated that sanguinarine exerted a weaker suppressive effect on mammalian cells than on *C. albicans* cells. These findings demonstrate that sanguinarine shows strong potential as an antimicrobial/antibiofilm agent for effective treatments of biofilm-associated infections caused by *C. albicans* and *S. aureus* mono- and mixed-species. However, the present study is only a preliminary investigation. Further in vivo and clinical investigations should be undertaken to verify the therapeutic beneficial effects of sanguinarine on treatment of polymicrobial infections.

## 4. Materials and Methods

### 4.1. Reagents

Sanguinarine (HPLC purity ≥ 98%) was purchased from Chengdu Pulis Biological Science and Technology Co., Ltd. (Chengdu, China). Sanguinarine was initially dissolved in dimethyl sulfoxide (DMSO) at a concentration of 40 mg/mL and filter-sterilized prior to use. SYTO 9, SYPRO Ruby, WGA, PI, and FM 4-64 dyes were obtained from Invitrogen (Thermo Fisher Scientific, Waltham, MA, USA). All other chemicals and solvents used in this study were of analytical grade.

### 4.2. Strains and Cultural Conditions

*C. albicans* SC5314 and *S. aureus* CMCC26003 were routinely incubated in yeast extract peptone dextrose (YPD), containing different concentrations of sanguinarine when necessary, at 37 °C in a shaking incubator, respectively. The fungal and bacterial stocks were stored at −70 °C supplemented with 25% glycerol as a cryoprotectant. To achieve biofilms of single species, overnight cultures of *C. albicans* SC5314 and *S. aureus* CMCC26003 were diluted 1:100 into fresh YPD and cultured at 37 °C until a concentration of approximately 1 × 10^6^ colony forming units (CFU)/mL was reached. Then, 1 mL of fresh cultures were transferred into individual wells of 24-well plates (Nunc, Copenhagen, Denmark) and sanguinarine was added into each well with final concentrations of 2–128 μg/mL when necessary. The plate was then incubated at 37 °C under anaerobic conditions for 24 h. To generate biofilms of dual species, overnight cultures of *C. albicans* SC5314 and *S. aureus* CMCC26003 were adjusted to an OD_600_ of 1.0 (approximately 1 × 10^8^ CFU/mL) with fresh YPD and an equal volume of each strain (100 μL) was added to prepared wells of the 24-well plate with a final volume of 1 mL.

### 4.3. Minimum Inhibitory Concentration (MIC)

The MIC values were measured for *C. albicans* SC5314 and *S. aureus* CMCC26003 according to the broth microdilution method and performed by a serial dilution technique using 96-well microtiter plates [36]. Briefly, each well was inoculated with 200 μL of inoculum containing approximately 1 × 10^6^ CFU/mL of bacteria and then final concentrations of sanguinarine ranging from 0 to 128 μg/mL were added to the pre-equilibrated medium in individual wells, respectively. Fluconazole (64 μg/mL) was employed as the positive control, while 1% DMSO was used as the negative group. The microplates were incubated at 37 °C for 24 h and the resulting samples were decimal diluted, plated onto YPD agar (YPDA) and incubated overnight at 37 °C to determine the number of CFU. The lowest concentration without visible growth was defined as the MIC.

### 4.4. Minimum Biofilm Inhibitory Concentration (MBIC90)

The minimum biofilm inhibitory concentration (MBIC90), which is defined as the lowest concentration of antimicrobials that results in the inhibition of 90% biofilm formation, was quantified by the crystal violet (CV) assay and measuring optical density (OD 570 nm) [37]. To generate single species biofilms, overnight cultures of *S. aureus* CMCC26003 mono-species were diluted 1:100 into fresh YPD, and 200 μL per well was deposited in 96-well microtiter plate. For the dual species biofilm assays, overnight cultures of *C. albicans* SC5314 and *S. aureus* CMCC26003 were normalized to an OD_600_ of 1.00 and an equal volume of each (50 μL) was added to prepared wells (final volume of 200 μL). Then, mono- and dual-species cultures were treated with sanguinarine at different concentrations (0, 1/16, 1/8, 1/4, 1/2, and MIC) and incubated at 37 °C for 24 h. After incubation, the biofilms formed in individual wells were fixed with 200 μL of methanol after being gently washed three times with 10 mM PBS. Then, 200 μL of 0.1% (*w*/*v*) CV dyes was added to each well, and incubated at room temperature for 15 min. The unbound free CV was removed and washed gently with distilled water three times, and 200 μL of 95% (*v*/*v*) ethanol was added to dissolve the CV that was combined with the biofilms. For each sample, the absorbance at 570 nm was recorded by a microplate reader (Thermo Fisher Scientific, Ratastie 2, Vantaa, Finland).

### 4.5. Biofilm Inhibition Assay

The inhibitory effects of sanguinarine on the biofilm formation were evaluated qualitatively by FESEM and CLSM with a few modifications [38,39]. Mono- and dual-species of *C. albicans* SC5314 and *S. aureus* CMCC26003 inoculum at approximately 1 × 10^8^ CFU/mL were cultured on 1 cm × 1 cm glass coverslips placed in individual wells of a 24-well plate, and exposed to sanguinarine at 0, 1/8, 1/4 and ½ MIC at 37 °C for 24 h, respectively. For FESEM analysis, mature biofilms were fixed in 2.5% glutaraldehyde (*v*/*v*) at −4 °C for 2 h and then rinsed three times with 0.01 mol/L phosphate buffer saline (PBS, pH 7.0). The samples were then dehydrated by washing with increasing amounts of ethanol: 30%, 50%, 70%, and 90% ethanol, each for 10 min, followed by 100% ethanol twice, each for 10 min. Finally, samples were examined under a FESEM. For CLSM analysis, mature biofilms were washed three times with 10 mM PBS to remove planktonic cells, and then stained with SYTO 9. After incubation at 25 °C for 15 min, the biofilms were examined by a CLSM, where the fluorescence was measured at excitation/emission wavelengths of 485/542 nm for SYTO 9.

### 4.6. Biofilm Composition by CLSM

The composition changes of 24-h biofilms of mono- or dual-species were observed by CLSM. Briefly, approximately 1 × 10^6^ CFU/mL of mono- and dual-species of *C. albicans* SC5314 and *S. aureus* CMCC26003 were cultured on the glass coverslip placed in each well and exposed to sanguinarine of a variety of concentrations (0, 1/8, 1/4, and ½ MIC) at 37 °C for 24 h. Then, the resulting samples were further exposed to the following five types of dyes: (I) SYTO 9 dye, which stains nucleic acids; (II) Film Tracer SYPRO Ruby biofilm matrix stain, which labels most classes of proteins; (III) WGA conjugated with Oregon Green, which stains polysaccharides; (IV) PI dye, that stains nucleic acids; and (V) FM 4-64 dye, which stains lipophilic membrane. The fluorescence of dyes was measured using the following combination of excitation and emission wavelengths: 476 nm/500–520 nm for SYTO 9, 405 nm/655–755 nm for SYPRO Ruby, 459 nm/505–540 nm for WGA, 535 nm/617–635 nm for PI, and 479 nm/565–588 nm for FM 4-64, respectively. After each staining step, the biofilms were gently rinsed with 10 mM PBS and observed under a CLSM using the oil-immersion objective. Red/green fluorescence ratios to assess extracellular matrix components, including proteins, polysaccharides, and eDNA were measured on SYPRO Ruby/SYTO 9, FM 4-64/WGA and PI/SYTO 9 images with KS 400 version 3.0 software (Carl Zeiss, Inc., Jena, Germany), respectively. The fluorescence area (in square pixels) was averaged for the images taken in two areas per biofilm from three independent biofilms.

### 4.7. Diffusion of Antibiotics within Biofilms

Antibiotic diffusion within biofilms was determined based on the intrinsic fluorescence of gatifoxacin by CLSM in order to evaluate the diffusion capability of antibiotics within *C. albicans* SC5314 and *S. aureus* CMCC26003 biofilms formed in the presence of sanguinarine [40]. We inoculated 1 mL of mono- and dual-species cell suspensions containing 10^6^ CFU/mL into each well of the 24-well plate and treated them with sanguinarine at final concentrations of 0, 1/8, 1/4, and ½ MIC at 37 °C for 24 h. After incubation, the biofilms were gently washed three times with 10 mM PBS and a final concentration of 0.4 mg/mL gatifoxacin was added and further incubated for 5 h at 37 °C. Next, to evaluate the gatifoxacin diffusion within biofilms, the fluorescent dye of SYTO 9 (3 μM) was applied to stain biofilms. Followed by incubation at 25 °C for 15 min, the samples were rinsed three times with 10 mM PBS to remove nonpenetrated gatifoxacin and observed using a CLSM. To assess the structure and size of the biofilms, a series of images in the z axis were obtained. At least three random fields were visualized for each biofilm, and representative images are presented.

### 4.8. Biofilm Eradication Assay

To assess the capability of sanguinarine in eliminating preformed biofilms of mono- and dual-species of *C. albicans* SC5314 and *S. aureus* CMCC26003, the biofilm eradication assay was performed in a 24-well microtiter plate, as previously described. Monoculture and mixed cultures with a 1:1 ratio were cultured for 36 h on the glass coverslip placed in each well, and then treated with sanguinarine at 0, 8, 16, and 32 MIC at 37 °C for 5 h. Then, for quantitative analysis, the biofilm biomass was measured using the CV assay as described above. For qualitative analysis, the biofilms were observed by FESEM according to the protocol described above.

### 4.9. Effect of Sanguinarine on C. albicans Mature Hypha

The influence of sanguinarine on the mature hypha of *C. albicans* SC5314 was assessed by FESEM. *C. albicans* SC5314 was initially cultured in YPD medium at 37 °C for 16 h in a shaking incubator (180 rpm), washed twice with 10 mM PBS and diluted to a concentration of approximately 1 × 10^6^ CFU/mL with serum-free RPMI 1640. Then, each well of a 24-well culture plate was seeded with 100 µL of cell dilutions in 900 µL of fresh RPMI 1640 medium supplemented with 10% FBS. The plates were placed in a humidified-controlled incubator with 5% CO_2_ at 37 °C for 36 h. Subsequently, the resulting cultures were exposed to sanguinarine at final concentrations of 0, 1/2, 1, and 2 MIC at 37 °C for 4 h, respectively. After incubation, the samples were observed using a FESEM.

### 4.10. Statistical Analysis

Each experiment was repeated at least five times, and data are shown as mean ± standard deviation (SD) from three independent experiments. Statistical analyses were performed with SPSS software (SPSS 8.0 for Windows). Analysis of variance (ANOVA) was carried out to examine any significant differences (*p* ≤ 0.01).

## Figures and Tables

**Figure 1 pharmaceuticals-13-00013-f001:** Inhibitory effects of different concentrations of sanguinarine on *C. albicans* SC5314 and *S. aureus* CMCC26003 mono- and dual-species biofilms by the crystal violet assay. Values represent the means of triplicate measurements. Bars represent the standard deviation (N = 6). * *p* < 0.05; ** *p* < 0.01; *** *p* < 0.001; NS, not significant.

**Figure 2 pharmaceuticals-13-00013-f002:** Examination of inhibitory effects of sub-MIC sanguinarine on *C. albicans* SC5314 (CA) and *S. aureus* CMCC26003 (SA) mono- and dual-species (SA+CA) biofilms by field emission scanning electron microscopy (FESEM, **A**) and confocal laser scanning microscopy (CLSM, **B**). Scale bars represent 5 μm for FESEM and 10 μm for CLSM, respectively.

**Figure 3 pharmaceuticals-13-00013-f003:** Evaluation of the effect of sub-MIC sanguinarine on *C. albicans* SC5314 (CA) and *S. aureus* CMCC26003 (SA) and dual-species (SA + CA) biofilm matrix structure by confocal laser scanning microscopy. (**A**) Extracellular proteins were stained with a red fluorescent stain SYPRO Ruby, and cells were stained with a green-fluorescent nucleic acid counterstain STYO 9. Scale bars represent 10 μm. (**B**) The median fluorescence intensity of extracellular proteins in biofilms of each treatment group was determined and plotted against that in untreated group by measuring red/green fluorescence ratios using KS 400 version 3.0 software. Bars represent the standard deviation (N = 6). *** *p* < 0.001.

**Figure 4 pharmaceuticals-13-00013-f004:** Investigation of the influence of sub-MIC sanguinarine on the levels of extracellular polysaccharides within *C. albicans* SC5314 (CA) and *S. aureus* CMCC26003 (SA) and dual-species (SA + CA) biofilms by confocal laser scanning microscopy. (**A**) Extracellular polysaccharides and microbial cells were labelled with a green fluorescent stain wheat germ agglutinin, and a red lipophilic membrane dye FM 4-64, respectively. Scale bars represent 10 μm. (**B**) The relative fluorescence intensity of extracellular polysaccharides in biofilms of each treatment group was calculated and plotted against that in untreated group by measuring red/green fluorescence ratios using KS 400 version 3.0 software. Bars represent the standard deviation (N = 6). *** *p* < 0.001.

**Figure 5 pharmaceuticals-13-00013-f005:** Effect of subMIC sanguinarine on the eDNA levels within *C. albicans* SC5314 (CA) and *S. aureus* CMCC26003 (SA) mono- and dual-species (SA + CA) biofilms by confocal laser scanning microscopy. (**A**) eDNA and microbial cells were labelled with a red fluorescent dye PI, and a green-fluorescent nucleic acid counterstain STYO 9, respectively. Scale bars represent 10 μm. (**B**) The relative fluorescence intensity of eDNA in biofilms of each treatment group was calculated and plotted against that in untreated group by measuring red/green fluorescence ratios using KS 400 version 3.0 software. Bars represent the standard deviation (N = 6). *** *p* < 0.001.

**Figure 6 pharmaceuticals-13-00013-f006:** Representative confocal laser scanning microscopy images evaluating diffusion of gatifloxacin within *C. albicans* SC5314 (CA) and *S. aureus* CMCC26003 (SA) mono- and dual-species (SA+CA) biofilms formed in the presence of sub-MIC sanguinarine. Biofilms were stained with SYTO 9 for biofilms (green) and the intrinsic fluorescence of gatifloxacin (blue). Scale bars represent 10 μm.

**Figure 7 pharmaceuticals-13-00013-f007:** Eradication effects of different concentrations of sanguinarine on mature biofilms of *C. albicans* SC5314 and *S. aureus* CMCC26003 mono- and dual-species by crystal violet assay (**A**) and field emission scanning electron microscopy (**B**), respectively. The biofilm biomass was determined using the crystal violet assay. Values represent the means of triplicate measurements. Bars represent the standard deviation (N = 6). ** *p* < 0.01; *** *p* < 0.001. Each field of vision was magnified 10,000×. Scale bars represent 5 μm.

**Figure 8 pharmaceuticals-13-00013-f008:** The effect of sanguinarine on the mature hyphae of *C. albicans* SC5314. The mature hypha of *C. albicans* SC5314 was treated with indicated concentrations of sanguinarine for 5 h at 37 °C. Each field of vision was magnified 5000×. Scale bars represent 10 μm.

**Table 1 pharmaceuticals-13-00013-t001:** MICs and MBIC90 of sanguinarine against mono- and dual-species of *C. albicans* SC5314 and *S. aureus* CMCC26003.

Strain	MICs (μg/mL) of Sanguinarine	MBIC90 (μg/mL) of Sanguinarine
*C. albicans* SC5314	4	2
*S. aureus* CMCC26003	4	2
Dual species	8	4

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
