# Peer review of "Sanguinarine Inhibits Mono- and Dual-Species Biofilm Formation by Candida albicans and Staphylococcus aureus and Induces Mature Hypha Transition of C. albicans"

_pharmaceuticals, 2020, doi:10.3390/ph13010013_

Round 1
Reviewer 1 Report
In this article, Qian et al. provide new insights on the development of new agents to combat biofilm-associated infections caused by the bacterias. The results are interesting and I recommend it for publication, once following comments have been taken care of.
These relevant articles must be included (Kim et al., New Journal of Physics 16 (2014) 065024; Mittal et al., ACS Appl. Mater. Interfaces 2017, 9, 23, 19371-19379). Quality of figures is poor. Scale bars in Figures 2A, 7B and 8 are not visible. Figures 1 and 7A should be replaced with the better quality figures. Authors should mention in the introduction that C. albicans is referred to Candida albicans.Author Response
Please see the attachment.

Reviewer 2 Report
The manuscript does not so interesting for reading, introduction and discussion are not large, as well as results description. All experimental data were calculated from 3 repeats that is not enough for serious research. I suggest to enlarge repeats up to N=5 (minimal).
According to table 1 the MICs for microorganisms was 4 μg / ml and 8 μg / ml for their combination, while MICB for biofilm was two times lower. In contrast on Figure 1, we see that at 1/2 MIC all three biofilms presented and their values are above 0. Authors presented conflicting data.
The document contains many abbreviations and their decryption is not given almost everywhere; it is difficult to understand this abbreviations, please write first time description for all.
Figures with microscopic data have not Scale bars but legends mention them. For this microscopic data I have one more pretension. Today a lot of computer programs are developed for digital image processing, so images must to be processed and digital data for each dyes should be compared further. It means the calculation of % of each dye for a standard area. The average scatter chart from several experiments can characterise the effect veraciously.
In line 135 was written “a large number of extracellular proteins were seen and evenly embedded in the biofilm…” It is not clear which proteins type are stained by this dye. Is these proteins located in the cell membrane or forming the external matrix? No comments from authors in results and discussion sections present. For readers more detailed information about type of proteins involved in biomatrix formation is essential as well as more detailed information about eDNA molecules (further result in the manuscript).
It is not clear what was drawn on Figure 6. According to the method, a biofilm depth was scannedю Where are the top or bottom of slices and how the antibiotic administration was done? Why on the figure some slices are thick but another thin, for effect comparing they should be keep the same geometric size, but different density of course.
The wrong methodology was utilized for gatifloxacin effectiveness estimation, since on Fig.6 an impaired biofilm has been selected for the test. In practice the antibiotic may be used onto already formed biofilm. According to the authors data in Fig. 7 active concentrations of sanguinarine in such model are 16 and 32 MIC. Experiment on preformed biofilms is essential to add to the manuscript otherwise no any estimation about prospective of sanguinarine as drug can be done.
Authors conclude in discussion that These findings demonstrate that sanguinarine shows strong potential as an antimicrobial/antibiofilm agent. Please at first provide the data on preformed biofilms and calculate the effective concentration of sanguinarine. At second, the therapy effect will be accompanied with a number of side effects, and authors have overestimation. They should to describe the side effects and the doses for sanguinarine from literature and compare doses harmful for mammalian with proposed treatment doses. As a result, the therapeutic index will be calculated.
Round 2
Reviewer 2 Report
Authors' responses to my comments suit me